# Free-IgE as a Predictor of Responsiveness to Omalizumab in Oral Corticosteroid-Dependent Asthma Patients

**DOI:** 10.3390/ijms26072852

**Published:** 2025-03-21

**Authors:** Christian Domingo, Daniel-Ross Monserrate, Markus Ollert, Xavier Pomares, Carles Forné, Jorge del Estal, María José Amengual

**Affiliations:** 1Department of Pulmonary Medicine, Parc Taulí Hospital Universitari, Institut d’Investigació i Innovació Parc Taulí (I3PT-CERCA), Universitat Autònoma de Barcelona, 08208 Sabadell, Barcelona, Spain; drmonserrate@tauli.cat (D.-R.M.);; 2Clinical Research Division of Molecular and Clinical Allergotoxicology, Department of Dermatology and Allergy, Technische Universität München, 80802 Münich, Germany; 3Heorfy Consulting, 25007 Lleida, Spain; 4Department of Basic Medical Sciences, University of Lleida, 25198 Lleida, Spain; 5Pharmacy Department, Parc Taulí Hospital Universitari, Institut d’Investigació i Innovació Parc Taulí (I3PT-CERCA), Universitat Autònoma de Barcelona, 08208 Sabadell, Barcelona, Spain; 6Laboratory Department, Immunology Unit, Parc Taulí Hospital Universitari, Institut d’Investigació i Innovació Parc Taulí (I3PT-CERCA), Universitat Autònoma de Barcelona, 08208 Sabadell, Barcelona, Spain

**Keywords:** blood IgE, free IgE, severe asthma, biomarker, predictor capacity, omalizumab, clinical response

## Abstract

To date, no biomarkers have been found that are able to predict the clinical response to omalizumab. The aim of this study was to assess whether blood concentration of free Immunoglobulin E (IgE) can predict response to treatment with this monoclonal antibody. In a group of patients who were candidates for omalizumab treatment, forced spirometry and blood IgE were measured at entry and at each six-month visit, and free-IgE blood concentrations were measured at month 6. At month 18, the OMADORE protocol was applied. The complete follow-up lasted 30 months. Patients were considered responders if they met at least one of the following criteria: increase in forced expiratory volume in one second (FEV_1_) at the follow-up visit compared to baseline; reduction in corticosteroid dose at the last visit compared to baseline; reduction in omalizumab dose at the follow-up visit; a positive score on the composite index combining all three criteria. The biomarker used to predict treatment response was the free IgE value and the percentage of free IgE to total IgE measured at visit 1, after six months of omalizumab treatment. The percentage of responders varied according to the parameter used (FEV_1_, omalizumab, corticosteroid dose, and the composite index; 45.2%, 64.5%, 48.4%, and 77.4%, respectively). IgE blockade was around 97% both for the group as a whole and for the subgroups. There were no differences in free IgE values nor in the ratio of free IgE to total IgE between responders and non-responders. These results confirm that there is a group of patients who may benefit from the reduction/withdrawal of omalizumab. Determination of free IgE six months after initiation of omalizumab treatment does not discriminate between responders and non-responders.

## 1. Introduction

When an atopic individual is exposed to an antigen, the dendritic cells (macrophages located in the organism’s epithelium) internalize it, process it, and present it to a T lymphocyte via the type two major histocompatibility complex. In this process, the T-lymphocyte develops a Th2 or a Th1 profile. When the type of antigen is an allergen, the lymphocyte differentiates into a Th2 cell. In these subjects, exposure to high concentrations of allergens leads to a high concentration of serum IgE, which favors the appearance of allergic symptoms. Th2 cells are able to produce IL-4, which, in turn, promotes the synthesis of IgE by the B-lymphocytes [1].

The rationale for the use of omalizumab is based on the drug’s ability to block IgE, thus inhibiting part of the Th2 immune response and reducing recruitment and activation of effector cells of the inflammatory process, including eosinophils [2,3]. Because IgE effector cells, such as mast cells and basophils, are a source of pro-inflammatory chemokines, cytokines, and proteases, anti-IgE therapy has an anti-inflammatory effect in asthma and other diseases [4]. Omalizumab has been shown to produce a 99% fall in free serum IgE within two hours; a downregulation of IgE receptors on basophils, mast cells, and dendritic cells after three months; a reduction in the number of eosinophils in sputum and bronchial biopsies; a reduction in exhaled nitric oxide; and a downregulation of IgE production [2,5]. Some studies have demonstrated the drug’s antiallergic and anti-inflammatory properties, such as the reduction in circulating tissue and sputum eosinophils [6,7], sometimes due to an increase in eosinophil apoptosis and a decrease in granulocyte-macrophage colony-stimulating factor (GM-CSF, IL-2, and Il-13).

Omalizumab treatment cannot be monitored by total blood IgE, because the drug binds to IgE (two molecules of omalizumab to one molecule of IgE), thus slowing the elimination of trimers and examers from the blood. Moreover, omalizumab is an IgG1 with a half-life ranging between 21 and 23 days, while the half-life of IgE is around 2–3 days. As a result, there is a higher concentration of IgE in the blood of treated patients, even though most of it is inactivated. Last but not least, until quite recently, the fraction of free IgE in blood could not be measured in patients treated with omalizumab [8].

Twenty years after omalizumab was first marketed, no biomarker has been identified that is able to predict patients’ response to the drug. The Global Evaluation of Treatment Effectiveness (GETE) is still the most frequently used parameter. Due to omalizumab’s effect on the IgE blockade, it made sense to investigate whether the percentage of IgE blockade differed in responders and non-responders.

The purpose of this study of a cohort of corticosteroid-dependent allergic asthma patients treated for six months with omalizumab is to determine whether the fraction of IgE not bound to the drug can predict a patient’s treatment response.

## 2. Results

Thirty-one patients (21 women; 10 men) were recruited, and there were no losses to follow-up. Demographic and initial clinical data are summarized in Table 1.

The longitudinal follow-up results for the whole cohort are shown in Appendix A. The IgE concentration increased 2.7-fold from baseline at visit 1 and then fell progressively to 1.8 times the baseline level by the end of the follow-up. After six months of treatment with omalizumab, 97.7% of IgE was blocked, which was calculated as follows from the values shown in Appendix A: free IgE concentration at V1 (6.6 IU/mL) divided by Total IgE concentration in V0 (285 IU/mL). For the whole cohort, at visit 5, the dose of omalizumab fell by 18% and the dose of OCs by 59.6%.

The percentage of responders at Visit 5 according to the different criteria used is summarized in Table 2. Figure 1 shows the evolution of the treatment response criteria over the course of the follow-up visits. For the composite index including the three criteria, 77.4% of patients were classified as responders at visit 5.

Comparisons of all study variables between responders and non-responders at visit 5 are shown in Appendix A (according to the composite index) and Appendix A (according to the omalizumab dose criterion). The evolution of FVC, FEV_1_, total IgE, and omalizumab dose are shown in Appendix A. A description of the initial and final free and total IgE values and the evolution of spirometry values is shown in Table 3.

Using the composite index as the treatment response criterion, we observed that the OC dose showed numerical differences in the responder group (initial/end responders: 6.5/2.08 mg; *p* < 0.001; Appendix A) but no changes were observed in the non-responders (initial/end non-responders: 3.14/3.14 mg; Appendix A). In the responder group, the OC dose rose slightly in 33.3% and fell in 66.7%.

Regarding spirometry values, 70.8% of responders presented improvement in the FEV_1_ at visit 5 (*p* = 0.007, Appendix A), and 29.2% a deterioration. None of the non-responders presented an improvement in this parameter.

In responders, IgE concentrations were higher at baseline, but lower at visit 5.

Using the omalizumab dose as the treatment response criterion we observed a non-significant improvement in FEV_1_ in the responder group and a significant reduction in OC dose (7.6 mg/0.91 mg per day; *p* < 0.003) (Appendix A).

The remaining parameters (free IgE, total IgE, and the free IgE/total IgE ratio) did not reveal significant differences (Appendix A).

The same comparisons using free IgE at visit 1 and the free IgE/total IgE ratio were performed according to the different treatment response criteria at visit 5 and did not reveal any difference (Figure 2 and Figure 3).

### 2.1. Analysis of Ability of IgE to Predict Treatment Response

Neither free IgE nor the free-to-total IgE ratio were useful predictors of clinical response to omalizumab, with AUCs (95% CI) of 0.452 (from 0.227 to 0.678) and 0.470 (from 0.223 to 0.717), respectively.

### 2.2. Side Effects

No relevant drug-related side effects were observed regarding drug tolerance.

## 3. Discussion

Classically, to predict the clinical response to omalizumab, the Global Evaluation of Treatment Effectiveness (GETE) has been used. No new biomarkers have been identified as reliable predictors of response to treatment. Since the pharmacological effect of omalizumab is to block IgE, it made sense to assess whether the percentage of IgE blockade might be related to patients’ clinical responses. Changes in these biomarkers have not been evaluated in the past.

In this study, a high percentage of patients treated with omalizumab presented a notable clinical response. The parameters used to measure this response were objective (reduction of OCS or omalizumab doses, improvement in pulmonary function tests, or in a composite index including all three parameters). Almost 80% of the treated patients met the responder criteria, while 97.7% of the circulating IgE was blocked. As expected, blood IgE notably increased during the first months of treatment.

Generally speaking, the increase in total IgE levels can be explained by the prolongation of the half-life of IgE (2–3 days) by binding to omalizumab, which has an IgG1 structure (with a half-life of 23–26 days).

Previous results published by our group show that there is a group of patients who may benefit from omalizumab reduction/withdrawal. The OMADORE protocol showed that, after starting omalizumab treatment, the OC dose can be reduced as far as the patient is able to tolerate it. The overall omalizumab saving was slightly below 50%. Thirty-four percent of patients tolerated omalizumab withdrawal, 23% tolerated a decrease in the dose, and 43% did not tolerate the decrease. The saving was obtained without any relevant or severe exacerbations.

Finally, the determination of free IgE six months after initiation of treatment with omalizumab did not allow discrimination between responders and non-responders.

IgE and its high-affinity receptor (FcεRI) play a pivotal role in the pathophysiology of asthma and other allergic diseases. Within minutes of exposure to the allergen, FcεRI+ mast cells bearing IgE directed against the relevant allergen become activated and release a variety of mediators, chemotactic factors, and cytokines. Omalizumab, an anti-IgE monoclonal antibody, has been reported to be effective in reducing the systemic side effects of subcutaneous immunotherapy for inhaled allergen in pollen-induced allergic rhinitis, mild–moderate allergic asthma, and venom allergy [9]. Omalizumab has also proved effective for pre-treatment and concurrent treatment of venom immunotherapy when anaphylaxis had occurred during initial immunotherapy [9]. As for its effect in acute allergic reactions (especially anaphylaxis), one would expect the IgE blockade to have an immediate effect on allergic asthma as well, but this is not the case; omalizumab does not play a central role in the management of allergic exacerbations of acute clinical asthma.

Omalizumab, an anti-IgE antibody used to treat severe allergic asthma and chronic idiopathic urticaria, binds to IgE in blood or membrane-bound on B lymphocytes, but not to IgE bound to its high (FcεRI) or low (CD23) affinity receptors in sensitized patients, which are ready to capture the allergens.

It should be noted that IgE must bind to its surface receptors, and, therefore, the total number as well as the percentage of receptors bound to IgE molecules may play an important role as IgE alone. Dendritic cells in peripheral blood (the stage in the dendritic cell life cycle that immediately precedes recruitment into the lung) did not show an increased expression of FcεRI in atopic asthmatic subjects compared to healthy, non-atopic subjects. These similar levels of expression of FcεRI in peripheral blood dendritic cells from healthy and asthmatic subjects suggest that the local environment in the airway is responsible for the upregulation of surface FcεRI on airway dendritic cells in asthma. The results also suggest that the functional ability of FcεRI to bind IgE is controlled differentially in the atopic state [10]. Once dendritic cells enter the lung, local microenvironmental factors may be responsible for the greater expression of FcεRI reported for these cells in the asthmatic airway [10]. This may be caused by the effect of thymic stromal lymphopoietin (TSLP), an alarmine released by the damaged bronchial epithelium that has been shown to regulate the activity of dendritic cells [11].

The expression of FcεRI on other cells present at allergic reaction sites, such as eosinophils, suggests that its function goes beyond its role in type I hypersensitivity [10].

The increased receptor expression may be due to the inhaled corticosteroid treatment that the subjects with more severe disease were receiving. Corticosteroids may cause an increase in FcεRI expression in dendritic cells by changing the differentiated state of the cell [12,13,14]; the maximum expression of FcεRI is found in immature dendritic cells.

Omalizumab (Xolair^®^, Novartis, Basel, Switzerland) is a humanized monoclonal antibody that binds selectively to IgE and prevents it from binding to the FcεRI receptor in basophils and mast cells, thereby reducing the amount of free IgE available to trigger the allergic cascade and also achieving a high reduction in the expression of these receptors on the surface of mast cells and basophils [15]. Moreover, because of the binding of omalizumab to free IgE, blood IgE levels fall by between 96% and 99%, and, as a result of the depletion of free IgE, the plasma membrane FcεRI receptors of basophils are transferred to the cytoplasm and are not resynthesized. This effect, which may contribute to explaining the pharmacodynamic effect of omalizumab, does not occur immediately [16].

Omalizumab binds to circulating free IgE and thereby inhibits the binding of IgE to FcεRI. Since it does not bind IgE on the cell surface, it does not directly affect mast cells and basophils. The drug has been shown to have an effect on the inhibition of IgE synthesis in peripheral blood mononuclear cells, not only at the level of protein synthesis but also in the expression of coding IgE mRNA [17]. Finally, by binding to the domain Cε3 of IgE expressed by B cells—IgE that has never been released to the blood but which is anchored to the cell membrane through a M’domain and not bound to the FcεRI [15]—omalizumab favors the apoptosis of B cells.

Omalizumab has been shown to have some effect on eosinophilic inflammation [7] by regulating the number of eosinophils in the sputum as well as in the bronchial mucosa. This effect may also be delayed.

Last but not least, the calculation of the drug dose may be inaccurate. The dose is estimated on the basis of the patient’s weight and plasma IgE concentration. When a sensitized patient undergoes massive exposure to the allergen, IgE production may increase, and the concentration of omalizumab in the blood may decrease as days pass; so, the dose received by the patient at that time may be inadequate. As a result, omalizumab may block the acute release of some mediators related to anaphylaxis, but, in asthma, there are many other pathways.

Regarding side effects, the drug was very well tolerated, and no relevant drug-related side effects were detected.

## 4. Methods and Materials

### 4.1. Hypothesis

Free IgE blood concentration after six months of treatment with omalizumab may be a predictor of drug response in patients with severe asthma.

### 4.2. Purpose

To assess the ability of free IgE concentration to predict treatment response after six months of treatment with omalizumab in patients with severe asthma.

### 4.3. Setting

Corporació Parc Taulí (CPT) (Sabadell/Catalonia), an 850-bed university hospital.

### 4.4. Design

Single-center, prospective, observational study.

### 4.5. Population

Patients were recruited from the severe asthma unit of our hospital.

### 4.6. Inclusion Criteria

Age ≥ 18 years old.Airway reversibility (Forced expiratory volume in one second, FEV_1_, reversibility ≥ 12% and 200 mL).Positive skin-prick test or in vitro reactivity (Immunecap) plus allergic symptoms.Baseline IgE level ranking between 30 and 1500 IU/mL.Weight between 25 and 150 kg.Patient naive to biological treatments.Oral corticosteroid dependence due to asthma, defined either as a requirement of a mean daily dose of at least 5 mg per day prednisolone to maintain an FEV_1_ > 50% during a period of one year or more in addition to the best standard care (BSC) provided by GINA or as the need for repeated boosters (≥5/year with an accumulated oral corticosteroid (OC) dose ≥ 300 mg/episode) during the previous year.Patient eligible for omalizumab treatment.Provision of written informed consent.

### 4.7. Exclusion Criteria

Refusal to sign the written informed consent document.Hypersensitivity to omalizumab.Impossibility of regular attendance at the asthma unit for control and drug administration.

### 4.8. Treatment Response Definitions

Patients were considered responders if they presented at least one of the following criteria:FEV_1_ at the follow-up visit higher than FEV_1_ at visit 0.Dose of corticosteroids at visit 5 lower than the dose of corticosteroids at visit 0.Dose of omalizumab at visit 5 lower than the dose of omalizumab at visit 1.Positive score on the composite index.

### 4.9. Methods

Treatment protocol: Patients received the best standard care (BSC) following the GINA recommendations. Prior to starting omalizumab treatment, patients underwent a stabilization period of at least three months. The stabilization period was followed by a one-month run-in period, after which the protocol was initiated. The follow-up period of treatment with omalizumab lasted 30 months.

The protocol followed for reducing OC administration was as follows: daily dose was decreased by 2 mg/day; if the patient remained stable at the end of the two weeks, the daily dose was decreased by a further 2 mg for the following two weeks until a decrease of ≥5% in the FEV_1_ was observed. The steroid dose was then increased to the previous level and the process was repeated. When the steroid decrease failed for the third time, it was considered that the minimum required steroid dosage had been reached.

### 4.10. Study Protocol (See Figure 4)

Total IgE concentration was measured at entry and every six months. Free IgE during omalizumab treatment was measured at visit 1 (after six months of treatment with omalizumab) according to the technique described elsewhere [8]. Forced spirometry was performed at each visit. OC and omalizumab doses were recorded.

**Figure 4 ijms-26-02852-f004:**
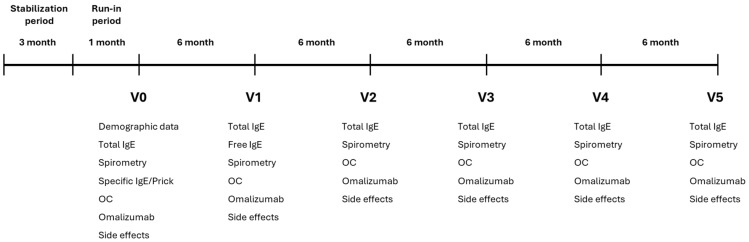
Study Protocol. OC: oral corticosteroids; IgE: Immunoglobulin E.

The omalizumab dose-decreasing protocol used was OMADORE, published by our group in 2018: to qualify for the step-down protocol, the patients had to have received treatment with omalizumab for at least 18 months: from month 0 to 6, the initial response to omalizumab treatment was assessed. From months 7 to 18, the omalizumab starting dose was maintained unchanged.

After 18 months, if the OC dose had reached the lowest tolerated dose (withdrawal or reduction) and the pulmonary function tests (PFTs) were greater than or equal to that at entry, the initial omalizumab dose was reduced by half. If clinically stable for 6 months, the dose was halved again. If needed, OC boosters were administered. When more than one OC booster was needed and/or PFTs worsened by ≥10%, the omalizumab dose was raised to the previous figure until the patient stabilized. The reduction in omalizumab dose was assessed every 6 months until withdrawal.

### 4.11. Informed Consent

Written informed consent was obtained from each patient. The protocol was approved by the internal review board of the hospital.

### 4.12. Side Effects

At each visit, patients were specifically asked about side effects. They were also given the telephone number of the outpatient unit of the pulmonary service and asked to report any suspected side effects.

### 4.13. Sample Size

There are no previous estimations of the required sample size in the literature. We considered that a cohort of 30 patients would be representative of a pilot study.

### 4.14. Statistical Analysis

Descriptive statistics were computed for each study variable at each follow-up visit. Continuous variables were summarized using means and standard deviations, and categorical variables were presented as frequencies and percentages.

Comparisons between responders and non-responders at visit 5 were conducted using the t test for continuous variables and the Pearson chi-square test for categorical variables. The paired t test was used to compare FEV_1_, OC dose, omalizumab dose, and total IgE between visit 5 and baseline in responders.

The predictive ability of free IgE and free-to-total IgE ratio at visit 1 for treatment response at visit 5 was assessed by calculating the area under the receiver operating characteristic (ROC) curve (AUC) along with its 95% confidence interval (CI).

### 4.15. Study Restrictions

As this is a pilot study, there is no previous information that would allow the calculation of the sample size. We, therefore, chose to evaluate a minimum number of 30 patients.

## 5. Conclusions

We conclude that a high percentage of patients treated with omalizumab (around 80%) present a good clinical response. The high rate of the IgE blockade obviously counteracts the situations triggered by the immediate effect of IgE release. The treatment causes an apparent increase in total IgE, but around 97% are inactive since they are bound to omalizumab. In asthma, the situation is different; apart from the immediate beneficial effect of the IgE blockade, many effects occur later, which may explain why the measurement of free IgE after six months of treatment does not distinguish between long-term responders and non-responders. Thus, the determination of free IgE does not allow discrimination between responders and non-responders and, therefore, is not a good biomarker for guiding modifications in the omalizumab dose designed to improve patients’ long-term clinical response. 

## Figures and Tables

**Figure 1 ijms-26-02852-f001:**
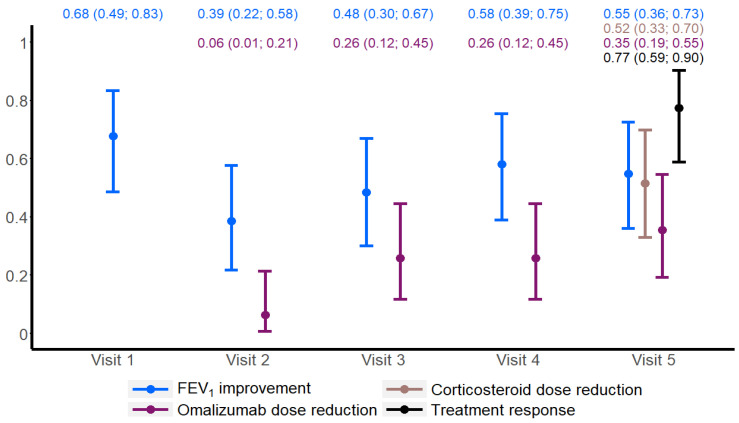
Evolution of the treatment response criteria throughout follow-up visits. This figure shows the percentages (95% confidence intervals) of responders at each visit.

**Figure 2 ijms-26-02852-f002:**
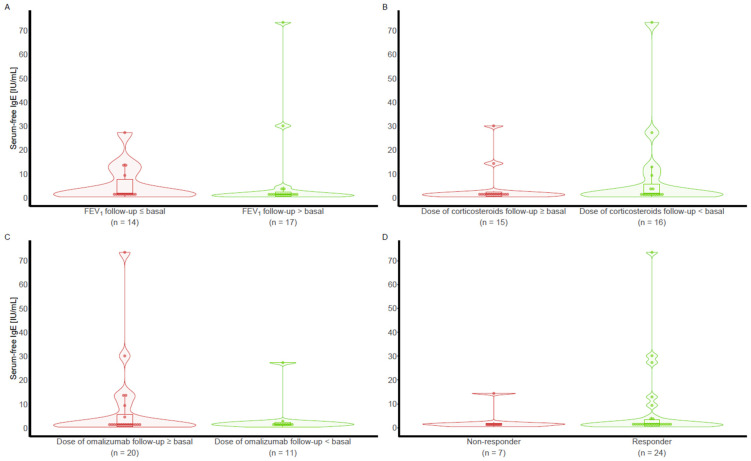
The concentration of free IgE according to the different treatment response criteria at visit 5:(**A**) FEV_1_; (**B**) corticosteroid dose; (**C**) omalizumab dose; (**D**) combined three criteria.

**Figure 3 ijms-26-02852-f003:**
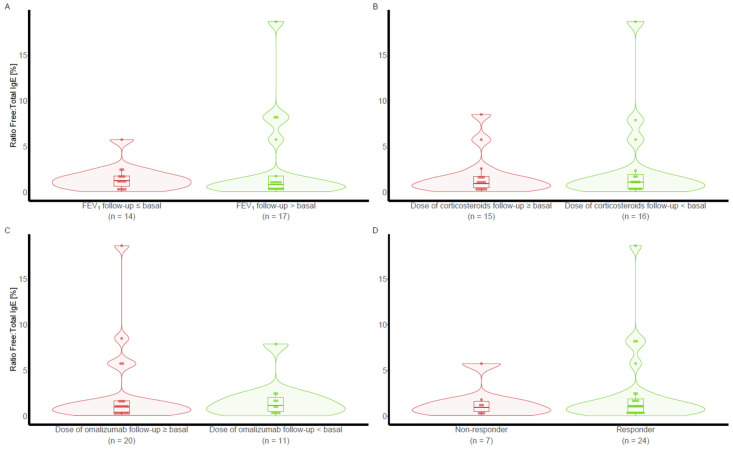
The concentration of the free IgE/total IgE ratio according to the different treatment response criteria at Visit 5: (**A**) FEV_1_ (**B**) corticosteroid dose; (**C**) omalizumab dose; (**D**) combined three criteria.

**Table 1 ijms-26-02852-t001:** Demographic and initial clinical data of the study cohort.

Variable	N = 31
Women	21 (67.7%)
Age at visit 0, years	51.7 (17.0)
Weight, kg	73.0 (17.9)
Height, cm	160 (9.8)
BMI, kg/m^2^	28.3 (6.0)
Total IgE, IU/mL	285 (280)
FVC, %	77.4 (20.1)
FEV_1_, %	63.5 (20.5)
FEV_1_/FVC ratio, %	62.9 (14.2)
Dose of corticosteroids, mg	5.7 (5.1)

Values are presented as means (standard deviation) for continuous variables and frequencies (percentage) for categorical variables. BMI, body mass index; FEV_1_, forced expiratory volume in 1 s; FVC, forced vital capacity; IgE, immunoglobulin E; IU, international unit.

**Table 2 ijms-26-02852-t002:** Percentage of responders/non-responders at visit 5 according to the different criteria of responsiveness applied.

	Treatment Response Criteria			
	FEV_1_	Oral Corticosteroids	Omalizumab	Three Criteria
Responders	17 (54.8%)	16 (51.6%)	11 (35.5%)	24 (77.4%)
Non-responders	14 (45.2%)	15 (48.4%)	20 (64.5%)	7 (22.6%)

FEV_1_, forced expiratory volume in 1 s.

**Table 3 ijms-26-02852-t003:** Description of the initial and final free and total IgE values and evolution of spirometry values.

	Three Combined Criteria		Omalizumab Dose Criterion	
	Responders	Non-Responders	Responders	Non-Responders
Total IgE at visit 0, IU/mL	302 (309)	228 (144)	310 (330)	271 (256)
Total IgE at visit 5, IU/mL	500 (462)	563 (417)	378 (219)	589 (522)
Free IgE at visit 1, IU/mL	7.62 (16.1)	3.21 (4.98)	4.01 (7.78)	8.06 (17.0)
FEV_1_ at visit 0, %	60.4 (20.6)	74.1 (17.0)	61.0 (17.0)	64.8 (22.5)
FEV_1_ at visit 5, %	68.3 (21.9)	58.3 (17.6)	68.9 (22.8)	64.4 (20.6)

Values are presented as means (standard deviation). FEV_1_, forced expiratory volume in 1 s; IgE, immunoglobulin E; IU, international unit.

## Data Availability

The original contributions presented in this study are included in the article/Appendix A. Further inquiries can be directed to the corresponding author.

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
