# Peer review of "Free-IgE as a Predictor of Responsiveness to Omalizumab in Oral Corticosteroid-Dependent Asthma Patients"

_ijms, 2025, doi:10.3390/ijms26072852_

Round 1

Reviewer 1 Report

Comments and Suggestions for Authors

Christian and colleagues presented study on Free-IgE as a predictor factor of responsiveness to omalizumab in oral corticosteroid-dependent asthma patients. They wanted to assess the predictive capacity of free IgE concentration after six months of omalizumab treatment in severe asthma patients. Although the hypothesis they want to test seems good, I found it hard to understand the content of the study as the methodology is not written appropriately and the results are not interpreted adequately.

I have the following specific comments.

1/ Abstract is not well structured, full of incomplete sentences with many abbreviations.

2/ Introduction doesn’t reflect the scientific rationale why they want to undertake this study.

3/ Methods are not written based on the recommended format. It lacks clarity and details.

4/ A lot of duplications in the materials and methods section

5/ Study design is confusing. What does Single-center, prospective, observational study mean?

6/ Sample size is too small for such cross-sectional study. This a limitation.

7/ Figure 1 is not illustrated the way it should be illustrated.

8/ A lot of definition is given for various statistical tools and why is it so important to describe one by one instead of putting in one paragraph?

9/ Figure 2 explains what? Why are error bars lager?

10/ All the statistical analysis in Figures 3 and 4 are not clear with not so much clarification and scientific interpretations.  

11/ A lot language edition, sentence reconstruction is required.

Comments on the Quality of English Language

Language work is required. 

Author Response

See word file attached. 

Reviewer 2 Report

Comments and Suggestions for Authors

The authors investigated whether the level of free IgE in patients' blood could be a biomarker of the efficacy of omalizumab treatment.

The manuscript needs improvement.

The information in the Abstract does not agree with the information in the Introduction. In the abstract the authors state that they studied patients treated for a minimum of 6 months, whereas in the Introduction they state that patients were treated for 6 months.

What were the side effects and how often did they occur? Was there a correlation between the severity of side effects and free IgE levels?

Please provide a more detailed description of Fig. 1 and Fig. 2. What is '0' on Fig.1?

Where are the E1  (mentioned in line 165) as well as E2 and E3 (mentioned in line 171) tables?

Table 1 needs to be corrected. The column " Valid N" should be removed and a more accurate table caption added. N=31 should be added in the table caption.

Where is the figure E1 (mentioned in line 201)?

The discussion must be improved.

What is the novelty of the study?

Comments on the Quality of English Language

Moderate editing of English language required.

Author Response

See word-file attached
